



# Long-term reduction in CO₂ emissions from the Elbe River due to water quality improvement

Mingyang Tian[1], Jens Hartmann[1], Gibran Romero-Mujalli[1, 2], Thorben Amann[1], Lishan Ran[3], Ji-Hyung Park[4]

[1]Institute for Geology, Center for Earth System Research and Sustainability (CEN), Universität Hamburg, Bundesstrasse 55, 20146 Hamburg, Germany
[2]Instituto de ciencias de la Tierra, Facultad de Ciencias, Universidad Central de Venezuela, Caracas, Venezuela
[3]Department of Geography, The University of Hong Kong, Hong Kong SAR, China
[4]Department of Environmental Science and Engineering, Ewha Womans University, Seoul 03760, Republic of Korea

*Correspondence to:* Mingyang Tian (mingyang.tian@studium.uni-hamburg.de)

## Abstract

Polluted rivers transport and transform large quantities of anthropogenically-derived organic carbon to coastal regions, and account for an unneglectable share of global $CO_2$ emissions. Effective river water management can enhance water quality and reduce $CO_2$ emissions from the surface water to the atmosphere. However, the effect of water management on future riverine $CO_2$ emissions remains a topic yet to be explored. In this context, the effects of water quality on riverine carbon dynamics were evaluated by examining the temporal variations in carbon fluxes from the Elbe River during a climate base period (around 30 years) from 1984 to 2018. The analysis of long-term data reveals that annual $CO_2$ emissions from the Elbe River have decreased from 3.8±1.7 Tg C yr$^{-1}$ in 1984 to 1.3±0.6 Tg C yr$^{-1}$ in 2018 (1 T = $10^{12}$), and the largest reduction occurred after the initial decade of reunification of Germany. The changes in wastewater treatment have largely reduced nutrient loads, mitigated eutrophication, impacted the quality of the transported carbon to the ocean, resulting in concurrent decreases in $CO_2$ emissions. The long-term trends in the Elbe River underscore the importance of water quality management for mitigating $CO_2$ emissions from polluted rivers around the globe.

## 1. Introduction

Rivers contribute a significant amount of $CO_2$ emissions into the atmosphere, equivalent to about 56–68 % of global anthropogenic $CO_2$ emissions according to the latest estimation (22.8–27.6 *vs* 40.5 Pg $CO_2$ yr$^{-1}$) (Liu et al., 2022; Battin et al., 2023; Friedlingstein et al., 2022), and this percentage continues to increase because the unprecedented anthropogenic stresses on riverine systems have led to many negative issues such as water pollution (Best, 2018; Ran et al., 2021), and polluted river with excessive amounts of organic carbon derived from human sources tend to release more $CO_2$ compared to their counterparts (Kempe, 1982; Yoon et al., 2017). With the growing concerns about climate change and the necessity for mitigation strategies, it is crucial to manage and reduce $CO_2$ emissions from rivers.

A viable strategy for reducing $CO_2$ emissions in polluted rivers could be enhancing the ecosystem's natural carbon absorption and storage capabilities through water quality management. This strategy involves curbing the influx of elements that are derived from human activities and are present in excessive quantities, such as organic carbon (OC), nitrogen (N), and phosphorous (P). These steps are crucial because oversaturated $CO_2$ from rivers is primarily originating from terrestrial organic carbon (OC) and the respiration of aquatic ecosystems (Cole et al., 2007; Raymond et al., 2013), which are influenced by water quality and nutrient availability (Yoon et al., 2017; Kim et al., 2021).



The reduction of OC through primary and secondary wastewater treatment can minimize the liable OC before discharging into rivers, thereby reducing $CO_2$ emissions via respiration processes (Begum et al., 2019; Soares et al., 2019). But the recovery of eutrophication by water quality management could be more nuanced due to phytoplankton behaviors. On the one hand, some studies suggest that eutrophic rivers can act as $CO_2$ sink because of plankton photosynthesis, especially in rivers with low flow velocity or lakes and reservoirs (Sun et al., 2021; Demars et al., 2016; Crawford et al., 2016). On the other

hand, increased phytoplankton-derived dissolved organic matter (DOM), can boost bacterial respiration(Begum et al., 2019; Kim et al., 2021). Overall, it seems recovery from eutrophication could also have the possibilities of both increasing or decreasing river $CO_2$ emissions.

The concept of Resource Use Efficiency (RUE) was introduced in this instance as an essential metric for assessing riverine $CO_2$ levels under different nutrient conditions. This is largely because the trophic status related to nutrient availability

significantly impacts the levels of $CO_2$ in rivers (Regnier et al., 2022; Kim et al., 2021). RUE is often defined as the proportion of available resources that is incorporated into biomass (Hodapp et al., 2019). The positive relationship between biodiversity and RUE has been based on a hypothesis of more efficient nutrient use in more diverse ecosystems, while the opposite relationship also observed in freshwater and coastal regions (Ptacnik et al., 2008; Frank et al., 2020; Filstrup et al., 2014; Connell, 1978).

Further, most existing studies focusing on $CO_2$ dynamics in contest of water quality improvement have targeted on short-term effects or only the partial pressure of $CO_2$ ($pCO_2$). Such fragmentary analyses fail to capture the coherence of long-term variations in $CO_2$ emissions, and analyzing $pCO_2$ alone is not comprehensive, as $CO_2$ efflux is determined by both $pCO_2$ and hydrological conditions. Therefore, we need continuous, long-term studies on $F_{CO2}$ in combination with aquatic carbon and nutrient dynamics in a river basin that has recovered from a previously polluted state. This would provide insights into the

$CO_2$ removal extent by water quality management and the underlying reasons for such removal.

In the 1980s, the Elbe River was one of the highest polluted rivers due to the pollutants input from industrial, agricultural, and municipal wastewater. Following Germany's reunification, a series of effective water management strategies led to improvements in water quality (Adams et al., 1996). Previous research conducted at "Hamburger Wasserwerke" station reported high $pCO_2$ (around 6000 ppm in the late 50s and early 60s) in the Elbe River from 1954-1977 (Kempe, 1982).

Additionally, a decreasing $pCO_2$ trend in the Elbe coastal region was observed from 1990s (from nearly 7000 µatm to around 2500 µatm) (Amann et al., 2012). However, comprehensive analysis and understanding of how $F_{CO2}$ efflux has changed in the entire Elbe River and the underlying mechanisms, based on long-term datasets, is still lacking.

In this study, a high-resolution $CO_2$ emission dataset in the Elbe River spanning from 1984 to 2018 is assembled. The time series is composed through the combination of historical water chemistry data from FIS-FGG (2023) and daily river flow

discharge from the GRADES (The Global Reach-scale A priori Discharge Estimates for SWOT)(Yamazaki et al., 2019). During this period, the Elbe basin experienced significant socioeconomic shifts across two periods. The initial period in the 1980s was characterized by severe pollution due to extensive anthropogenic activities, while the later period presented improved water quality because of effective pollution management strategies. Based on this dataset, the effects of water quality on riverine carbon dynamics are examined through an in-depth time series analysis. In addition, the fluxes of key

carbon components (dissolved organic carbon, DOC, particulate organic carbon, POC; and dissolved inorganic carbon, DIC) over the same period is also estimated for tracking the temporal evolution of carbon fluxes in the Elbe River. The region-specific findings hypotheses the considerable potential of water quality management as an effective strategy for mitigating



$CO_2$ emissions. This is particularly significant because only a small proportion of wastewater is treated before discharging to river networks today (UN Water, 2021).


## 2. Methods and Materials

### 2.1 Study area

The Elbe River is one of the largest rivers in central Europe, emanates from the southern foothills of the Krkonoše Mountains, transverses Bohemia in the northwest of the Czech Republic, and subsequently enters eastern Germany from Dresden. With a length of 1049 km, it eventually discharges into the North Sea at Cuxhaven (Fig. S1). According to the Köppen-Geiger climate classification system, the prevailing climate type can be classified as "Cfb", indicating a warm temperate climate, fully humid, and with warm summers (Geiger, 1954; Köppen, 1936). The annual mean air temperatures are 8–9 °C in the lowlands and 1–3 °C at the summits of the low mountain ranges. The mean annual precipitation of the whole Elbe River basin is 628 mm (IKSE, 2022). The land cover types were mainly cropland and forest regions, accounting for 59.3% and 28.8% of the total basin in 1990. In 2018, the cropland-covered area slightly decreased by 2.2% while the forest-covered area increased by 1.1% (Copernicus-Land-Montoring-Service, 2022).

The Elbe River basin exhibited an excess of nutrients and organic carbon resulting from extensive anthropogenic interventions since the 1950s (Kempe, 1982), which was also observed at the Elbe estuary and other European rivers during the same period (Kempe, 1988; Abril et al., 2002; Amann et al., 2012). Since 1990, the German government has implemented a set of water pollution prevention and treatment strategies to restore water quality. Closing factories and reducing production output in the new established German federal states mainly contributed to the improvements in Elbe water quality during 1990-1992. Furthermore, the advancement of agricultural structure also played a critical role in reducing the input of nitrogen and phosphorus from croplands (IKSE, 2022). Additionally, tertiary wastewater treatment plants (WWTPs) are increasing across the country and remove organic matter and nutrients through physical and biological processes before they enter rivers (Kirschbaum and Richter, 2014). At the same time, the Czech Republic also adopted some endeavors for reducing the input of nutrients and pollutants from the upper Elbe River to improve water quality after 1989 (Adams et al., 1996; Guhr et al., 2000; Langhammer, 2010).

### 2.2 Dataset and pre-processing

To explore the temporal variations of carbon and nutrient dynamics in the Elbe River, a water chemistry dataset with 10 parameters from the FIS FGG was selected, the original data could be downloaded from the official website (https://www.fgg-elbe.de). The dataset is based on fixed monitoring stations from the German/Czech border before the Elbe estuary (Fig. S1). The sampling covers from 1984–2018 and normally with a frequency of 1–7 times per month. Generally, the analytical methods were consistent with the standard methods of the German Institute for Standardization, and the accuracy and precision could be found in the report from FGG Elbe. The water quality data including pH, temperature, alkalinity, for $CO_2$ calculations. Total nitrogen (TN), total phosphorus (TP), Total organic carbon (TOC), dissolved organic carbon (DOC), dissolved oxygen in percentage (%DO), chemical oxygen demand (COD), for the water quality improvement parameters. The particulate organic carbon (POC) was calculated as the difference between TOC and DOC. Chlorophyll-a to reflect the biomass variations from 1984 to 2018. For these parameters, pH was typically measured using potentiometric electrodes, temperature and %DO were measured with electrometric probes. TOC and DOC were assessed through catalytic high-temperature oxidation, TN and Chlorophyll-a were determined photometrically. Alkalinity and COD were measured using



the titration method (FIS-FGG, 2023). While during this research, we focus on the TN, TP, TOC, DO as the indicator for water quality improvement and nutrient analysis, and other parameters for the regression analysis.

The original dataset also contained a description of data quality, and any values that were below or above the detection
limit, as well as negative or zero values, were eliminated before analysis. The final dataset contains a total of 74 stations and 140,042 samples. Besides, to reduce the effect of the extreme observations, the annual median value instead mean value of multiple stations was applied to include the spatial differences of mainstem sites during time series analysis. The trends were analyzed by Mann-Kendall test (Kendall, 1948; Mann, 1945), the change points of mean shifts were detected by the least squared deviation. The proximity of large cities to main streams renders them more susceptible to anthropogenic activities,
whereas tributaries tend to be less impacted. Therefore, according to Strahler stream orders provided from EU-Hydro-River Network Database Version 1.3 (EEA, 2022), we separated the tributaries (Strahler orders 1-5) and main streams (Strahler orders 6-8).

For the analysis the dynamic between nutrients and carbon, the resource use efficiency of nitrogen ($RUE_N$) and phosphorous ($RUE_P$) was calculated as the ratio of phytoplankton biomass (Chlorophyll-a concentration) to the resource
availability (TN, TP, concentration) (Ptacnik et al., 2008; Kim et al., 2021):

$$RUE = \frac{\text{Chlorophyll-a (mg m}^{-3})}{TN \ (mg \ L^{-1}) \ and \ TP \ (mg \ L^{-1})} \qquad (1)$$

### 2.3 Annual $CO_2$ efflux calculation

$CO_2$ emissions were estimated from historical data for water chemistry, hydrological, and geographic data because there
are no direct continuous measurements of $CO_2$ efflux in the Elbe. The annual $CO_2$ emissions from the Elbe were calculated by daily $CO_2$ efflux ($F_{CO2}$) and water surface area ($S_w$), and the errors of these efflux calculations were tested using a Monte Carlo simulation with 1000 iterations (Supplementary Sections 1):

$$F_{CO2total}(\text{Tg C } yr^{-1}) = F_{CO2} \ (\text{mmol } m^{-2} \ d^{-1}) \times S_w \ (km^2) \qquad (2)$$

The $F_{CO2}$ was calculated by partial pressure of $CO_2$ in rivers ($pCO_{2 \ river}$) and atmospheric ($pCO_{2 \ air}$), the normalized
gas transfer velocity with a Schmidt number of 600 ($k_{600}$), and the Henry's constant for $CO_2$ corrected for temperature and pressure ($k_H$):

$$F_{CO2}(\text{mmol } m^{-2} \ d^{-1}) = [pCO_{2 \ river}(\mu atm) - pCO_{2 \ air}(\mu atm)] \times k_{600}(\text{cm } h^{-1}) \times k_H \ (mol \ L^{-1} \ atm^{-1}) \qquad (3)$$

The $pCO_{2 \ river}$ was estimated by the CO2SYS via pH, water temperature (T), and alkalinity (Lewis et al., 1998). The pCO2 results with the pH ≥ 6.5 were retained for the analysis because of the huge overestimation in low pH and ion strength
conditions (Abril et al., 2015; Nayna et al., 2021), and it is notable to indicate that the calculation methods using alkalinity could cause more uncertainties compared with dissolved inorganic carbon (DIC) (Romero-Mujalli et al., 2019) (Supplementary Section 1).

$k_{600}$ was calculated by equation (5) of Raymond et al. (2012) via channel slope (S) and flow velocity (v) as follow:

$$k_{600} = 2841 \times S \ (unitless) \times v \ (m \ s^{-1}) + 2.02 \qquad (4)$$

The v was scaled by the mean of two equations derived from world rivers via flow discharge (Q) (Raymond., et al., 2012):

$$\ln v \ (m \ s^{-1}) = -1.64 + 0.285 \times \ln Q \ (m^3 \ s^{-1}) \qquad (5)$$



$$\ln v \ (m \ s^{-1}) \ = -1.06 + 0.12 \times \ln Q \ (m^3 \ s^{-1}) \qquad (6)$$

The $Q$ for each station was extracted from the global reach-scale a priori discharge estimates for SWOT (GRADES), which were derived from MERIT-Hydro (Yamazaki et al., 2019), and covered the period from 1979–2019 (Lin et al., 2019; Yang et al., 2019). The $S$ for each station was scaled by QGIS 3.26.2 based on three datasets: the MERIT-Basins vector hydrography dataset (Lin et al., 2019; Yamazaki et al., 2019), EU-Hydro-River Network Database Version 1.3, and the EU-DEM v1.0 dataset (EEA, 2022). Annual mean atmospheric $pCO_2$ data were obtained from the NOAA Global Monitoring Laboratory (NOAA, 2022).

$S_w$ of the Elbe River was scaled through river length and river width reference to three products of the river network of EU-Hydro, HydroRIVERS (Lehner and Grill, 2013), and MERIT-Basins. The annual flow discharge was resampled from the daily GRADES discharge, in combination with the river length from the MERIT-Basins dataset. The river width was estimated by the follow equation (Raymond., et al., 2012):

$$\ln width \ (m) = 2.56 + 0.423 \times \ln Q \ (m^3 \ s^{-1}) \qquad (7)$$

## 2.4 Estimation of carbon fluxes

The fluxes of DIC, DOC, and POC were scaled from 1990s to 2018 in the Elbe River from Geesthacht station, as a basis for comparison with $CO_2$ efflux. The carbon loads to the Elbe estuary were calculated through two curve-fitting methods. The first method was based on the segmented log-log relationship between concentration ($C$) and instant flow discharge ($Q$) (Bakhmeteff, 1912). The datasets were segmented into two parts by a truncated exponent of median Q, and afterward, define C-Q patterns separately as $b_{50inf}$ when $Q < Q_{median}$, and $b_{50sup}$ when $Q > Q_{median}$ (Meybeck and Moatar, 2012):

$$C = a \times Q^{b50inf} \ (m^3 \ s^{-1}) \qquad (8)$$

$$C = a \times Q^{b50sup} (m^3 \ s^{-1}) \qquad (9)$$

The second estimation was calculated by a FORTRAN program software of Load Estimator (LOADEST) developed by the U.S. Geological Survey (USGS) (Runkel et al., 2004). The LOADEST contains 9 models including the log-log relationship but does not consider the segment between high and low flows, and the software can automatically select the best model according to the residual values (Supplementary Sections 2).

The $Q$ from the neighborhood hydrological station (Neu Darchau) was applied for the calculation because direct $Q$ measurements in Geesthacht station are not available. The daily $Q$ of 10 stations along the Elbe from Schöna (km 2.1) to Neu Darchau (km 536.44), which were provided by the German Federal Waterways and Shipping Administration (WSV), communicated by the German Federal Institute of Hydrology (BfG). Besides, due to the paucity of continuous DOC, POC, and DIC data in the Geesthacht station from 1984–2018, we join the concentration data from adjacent stations (Zollenspieker and Schnackenburg) for loads estimation to the Elbe estuary. DIC concentration was also calculated by the CO2SYS via pH, water temperature (T), and total alkalinity.

To identify the seasonal signal and extract robust load trends for each component through time, the seasonal-trend decomposition was applied using LOESS (STL) for the monthly loads, separated time series to trend, seasonal, and remainder (Cleveland et al., 1990). For $v = 1 \ to \ N$, then

$$Y_v = T_v + S_v + R_v \qquad (10)$$



Where, $Y_v$, $T_v$, $S_v$, and $R_v$ represent the data, the trend component, the seasonal component, and the remainder component, respectively.

190

## 3   Results

### 3.1   Long-term water quality improvement in the Elbe River

From 1984 to 1990, the Elbe River's mainstem had relatively high TN concentrations, averaging around 529±50 µmol L$^{-1}$ annually. However, a continuous decline has been observed since then, with the annual average decrease by about 12 µmol L$^{-1}$, reaching 220±28 µmol L$^{-1}$ in 2018. Conversely, the TN concentrations in the tributaries began to decline gradually after 1995 and remained stable within 207±92 µmol L$^{-1}$ up to 2018 (Fig. 1a).

Both the mainstem and tributaries of the Elbe River exhibited high TP concentrations until 1990 (17±4 µmol L$^{-1}$ in the mainstem, 12±2 µmol L$^{-1}$ in the tributaries). The most considerable decline in the mainstem occurred between 1990 and 1995, with an annual average decrease of 1.7 µmol L$^{-1}$. Another notable decrease happened between 1996 and 2005, reaching a relatively stable TP concentration of 5±1 µmol L$^{-1}$ by 2005 that remained unchanged until 2018. The tributaries similarly showed a gradual reduction in TP concentrations, with the greatest decrease observed before 1994 at an annual mean decline of approximately 1.5 µmol L$^{-1}$. From 1999 to 2018, TP concentrations in the tributaries remained relatively low, averaging around 4.0±0.4 µmol L$^{-1}$ (Fig. 1b).

Prior to 1990, the TN/TP ratio in the mainstream was relatively low, averaging 30±3 in 1984. This ratio then embarked on an upward trend, peaking at approximately 73±16 in 2013, afterward reversing into a steady decline, resulting in an annual mean of 50±9 by 2018. In contrast, the tributaries exhibited a more stable pattern for the TN/TP ratio starting in 1994, with an annual average of 76±38 and a specific mean of 74±36 in 2018.

Regarding TOC, a gradual reduction was noted in the mainstem from 1991 to 2018, with the most prominent decrease occurring before 1996 (a 11% annual decrease compared to pre-1990 levels, 1124±123 µmol L$^{-1}$). After 1996, the mainstem's TOC concentration fluctuated until 2018, averaging 660±89 µmol L$^{-1}$. In contrast, the tributaries' TOC concentration remained relatively stable between 1993 and 2018, averaging 610±169 µmol L$^{-1}$ (Fig. 1d).

%DO showed different patterns in the mainstem and the tributaries of the Elbe River. The tributaries displayed relatively small changes in %DO, with averages of 82±9% before 1990 and 92±9% in 2018. On the other hand, the mainstem had a lower %DO until 1990 (68±11%), but a gradual increasing trend was seen after 1990, reaching parity with the tributaries by 1992. The %DO in the mainstem continued to rise thereafter, consistently surpassing the levels in the tributaries, and reaching 102±11% in 2018 (Fig. 1e).

The results showed that the mainstem experienced a greater decrease in comparison to the tributaries. To investigate the longitudinal variations, the mainstem section was divided into three parts, each with an average distance of 200 km starting from the Czech-German border (Fig.S2). All three river segments showed a decreasing trend in the three parameters over time, with the upper region exhibiting the highest TN concentration from 1990 to 2018, followed by the middle and lower. Similarly, for TP, a decreasing trend was observed from upstream to downstream after 1992. Although little trend was observed in TOC for the upper, middle, and lower reaches of the mainstem, all three reaches demonstrated a consistent decreasing trend. %DO in the three river segments exhibited an increasing trend after 1993, showing a similar decreasing concentration trend from upstream to downstream over time, as observed for TN and TP (Fig. S2).





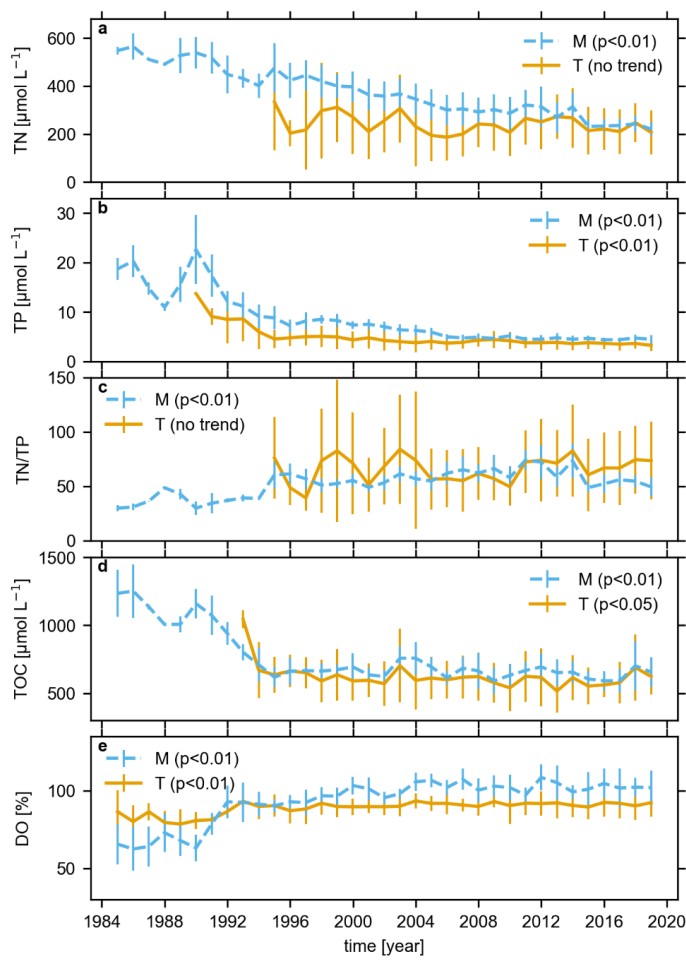

**Figure 1 Long term trends in water quality in the Elbe from the 1990s to 2018: (a) TN, (b) TP, (c) TN/TP, (d) TOC, (e) %DO. Values are annual mean with standard deviation indicating spatial variations within mainstem (M, n = 22) and tributary (T, n = 37). The red/blue dots represent the mainstem/tributary, dashed line means data gap. The *p* value indicated the significance level of each trend after passing through the M-K test.**

### 3.2 Long-term CO₂ emission changes in the Elbe River

From 1984 to 2018, the Elbe River primarily acted as a source of $CO_2$, with the $pCO_2$ levels ranging from 9 to 26,813 µatm (mean of 2290 µatm and median of 1631 µatm), while 13% (1235 of 9483) of the $pCO_2$ values were below the corresponding atmosphere level during the study period (mean of 381±15 µatm). Before 1990, the Elbe River presented high $pCO_2$ levels with an annual mean of 7185±2475 µatm. After 1990, $pCO_2$ began to decrease with the restoration efforts by the FGG Elbe, while the largest decrease occurred from 1991–1999 with an annual mean of 2259±1246 µatm. From 2000–2018, the annual mean $pCO_2$ stabilized at around 1797±1572 µatm, and the annual mean $pCO_2$ in 2018 was at 1832±1442 µatm (Table S1).

Subsequently, we integrated the historical $pCO_2$ and $k_{600}$ to determine the $F_{CO2}$ of the surface water area of the Elbe River catchment above Geesthast station. The $F_{CO2}$ of the Elbe River catchment varies from –30 to 2204 mmol m⁻² d⁻¹ with a mean

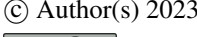



of 156±207 mmol m$^{-2}$ d$^{-1}$ from 1984 to 2018. Like the trend in pCO$_2$, the highest annual F$_{CO2}$ was observed pre-1990 (588±219 mmol m$^{-2}$ d$^{-1}$), with the largest decline occurring in 1999 (99±63 mmol m$^{-2}$ d$^{-1}$). The F$_{CO2}$, then remained relatively stable from 2000–2018 (116±132 mmol m$^{-2}$ d$^{-1}$).

Elevated F$_{CO2}$ was observed in both mainstem and the tributaries before 1990, with a mean of 599±276 mmol m$^{-2}$ d$^{-1}$ and 530±143 mmol m$^{-2}$ d$^{-1}$, respectively (Fig. 2a, Table S2). The highest F$_{CO2}$ in the mainstem and tributaries were observed at

station "Boizenburg (Strom-km 559,0)" on 1984-09-10 (2087 mmol m$^{-2}$ d$^{-1}$) and station "Boizenburg (Boize)" on 1987-11-17 (2204 mmol m$^{-2}$ d$^{-1}$). Following 1990, the frequency and magnitude of F$_{CO2}$ in the Elbe River started declining gradually, particularly in the mainstem. The maximum F$_{CO2}$ values in both the mainstem and tributaries after 1990 were detected at station "Zehren, rechtes Ufer (Storm-km 89,6)" on 1997-02-04 (1114 mmol m$^{-2}$ d$^{-1}$) and site "Gorsdorf (km 3,8)" on 2004-08-12 (2040 mmol m$^{-2}$ d$^{-1}$), respectively.

The lowest F$_{CO2}$ in the mainstem was observed around 2003, with an annual mean of 21±26 mmol m$^{-2}$ d$^{-1}$. From 2004 to 2007, the overall F$_{CO2}$ maintained a relatively low range (42±38 mmol m$^{-2}$ d$^{-1}$), although during this period there was a slight increasing trend in F$_{CO2}$. The annual mean F$_{CO2}$ was 59±35 mmol m$^{-2}$ d$^{-1}$ between 2008 and 2018. The F$_{CO2}$ also gradually increased longitudinally along the mainstem from upstream to downstream. However, the lower reaches of the river displayed lower F$_{CO2}$ values than the upstream and midstream reaches (Fig. S2). In the Elbe tributaries, there was a decrease in F$_{CO2}$

when compared to levels before 1990. Nevertheless, it is important to highlight that the F$_{CO2}$ in the tributaries remained higher than that in the mainstem of the Elbe River particularly after 1994, with the F$_{CO2}$ 2.4 times higher in the tributaries than in the mainstem (Fig. 2a, Table S2).

Both the mainstem and the tributaries exhibited notable seasonal fluctuations in F$_{CO2}$. The highest median F$_{CO2}$ in the mainstem was observed in December (113±170 mmol m$^{-2}$ d$^{-1}$), while in the tributaries, the highest median F$_{CO2}$ was observed

in August (277±165 mmol m$^{-2}$ d$^{-1}$). Conversely, the lowest median F$_{CO2}$ was found in May for the mainstem (61±190 mmol m$^{-2}$ d$^{-1}$) and in September for the tributaries (219±166 mmol m$^{-2}$ d$^{-1}$) (Fig. 2c).

The linear regression analysis was performed using data from the mainstem (Fig. 3). TOC showed significant positive relation with COD (r$^2$=0.44, p<0.01). %DO and RUE showed significant negative relation with lgpCO$_2$, (%DO, r$^2$=0.55, p<0.01) (RUE$_N$, r$^2$ = 0.50, p<0.01) (RUE$_P$, r$^2$=0.46, p<0.01), while COD showed significant positive relation with TOC

(r$^2$=0.7, p<0.01). We also conducted correlation analyses for pCO$_2$, k$_{600}$, and F$_{CO2}$, and the results revealed that pCO$_2$ had a strong correlation with F$_{CO2}$ (r$^2$=0.99, p<0.01), whereas k$_{600}$ and alkalinity did not show significant correlation with F$_{CO2}$.





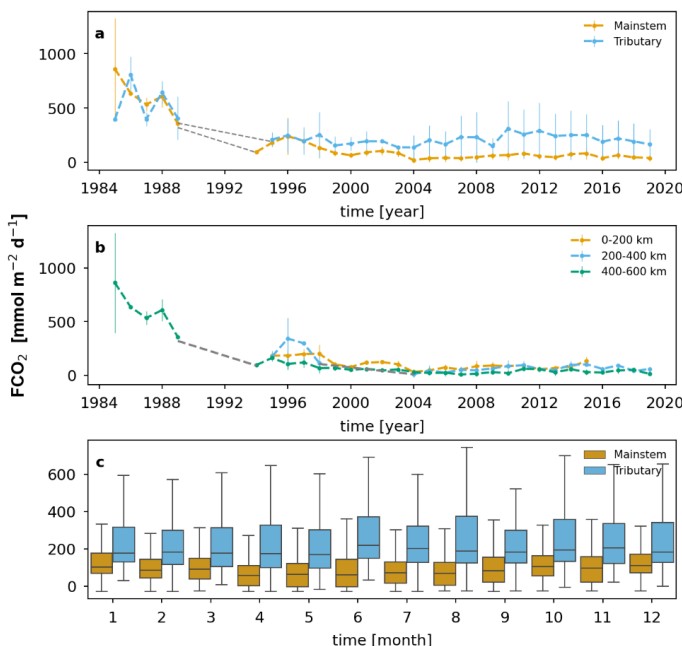

**Figure 2. Long-term trends in annual estimation of $F_{CO_2}$, values are annual means with standard deviation within each group: (A) mainstem and tributary of the Elbe River; (B) Up (0-200 km), Mid (200-400 km), and Down (400-600 km) reaches of the mainstem, number indicated the distance to the boundary of Germany/Czech. (C) seasonal variation of $F_{CO_2}$ within mainstem and tributary for the Elbe River from the 1990s to 2018**




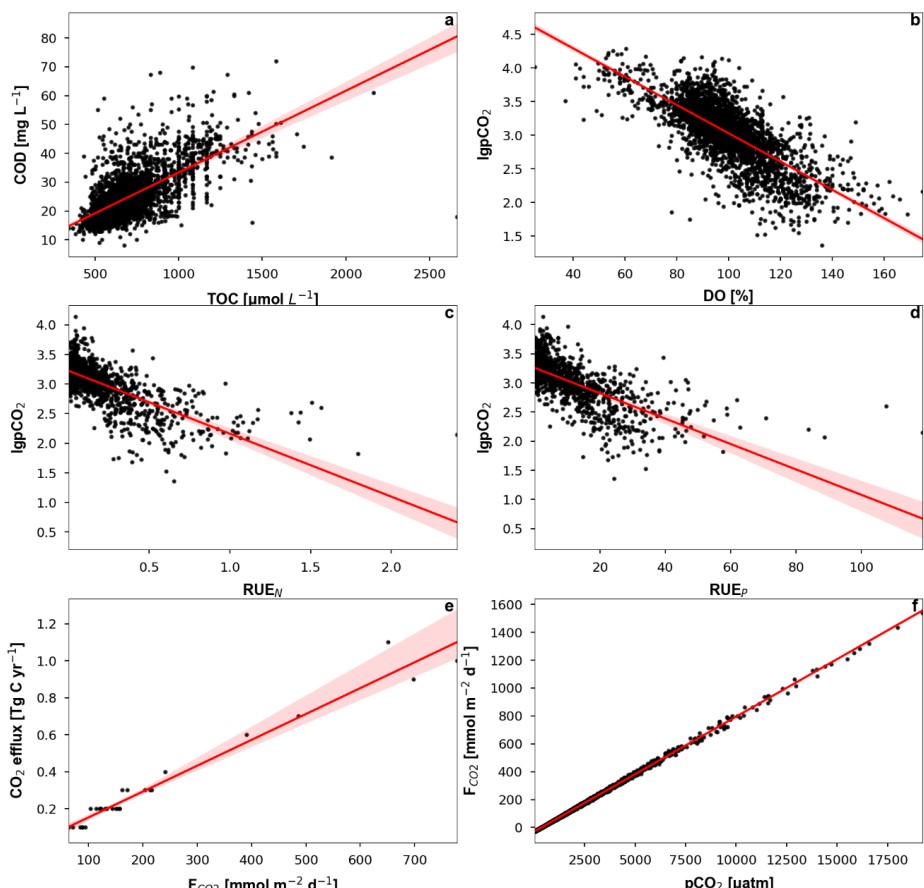

**Figure 3, the correlation between (a) TOC µmol L⁻¹ and COD mg L⁻¹; (b) %DO and lgCO₂; (c) RUEₙ and lgpCO₂; (d) RUEₚ and lgpCO₂ (e) FCO₂ mmol m⁻² d⁻¹ and F_CO2total Tg C yr⁻¹; (f) pCO₂ µatm and FCO₂ mmol m⁻² d⁻¹.**


### 3.3 Temporal variations in carbon fluxes in the Elbe River

In combination with annual water surface area (Table S1), the results show that areal $CO_2$ emissions from the Elbe remain at 1.3±0.6 Tg C yr⁻¹ in 2018 and much lower than that in 1984 (3.8±1.7 Tg C yr⁻¹), but it still equivalents to 3.5 times the corresponding DIC loads (0.37±0.10 Tg C yr⁻¹) to the Elbe estuary (Fig. 4a). The annual mean reduction amounts to about

0.08 Tg C yr⁻¹ from 1990s to 2018, with the most substantial decrease taking place in the initial decade, which corresponds to the elimination of 0.22 Tg C yr⁻¹ by water quality management. In contrast, POC, DOC, and DIC loads did not showed significant trends: In 1984, the annual POC, DOC, and DIC loads were 0.05±0.05, 0.21±0.04, and 0.67±0.10 Tg C yr⁻¹, respectively, while the values in 2018 were 0.04±0.07, 0.08±0.01, 0.37±0.10 Tg C yr⁻¹, respectively. (Fig. 4b, Table S1).

The predominant type of carbon discharged into the estuary was DIC, accounting for more than 70% of the total

(DIC+DOC+POC), the flux of DIC experienced a slight increase after 1990 and remained relatively stable at around 75% by 2018. Prior to 1990, DOC comprised roughly 23% of the carbon fluxes, but after 1990, its proportion decreased to around 17% and remain stable until 2018 (Fig. 4c). The lowest proportion of the three carbon types was POC, which was around 5%



in 1990. Between 1990 and 2018, the POC proportion slightly increased, and as of 2018, it accounted for approximately 8% of the three carbon fluxes. Nonetheless, compared to the $CO_2$ efflux it remains a relatively minor order of magnitude. Before 1990, the $CO_2$ efflux was 5.5 times greater than the DIC input to the coastal area, and as of 2018, this proportion still stands at 3.5 times.

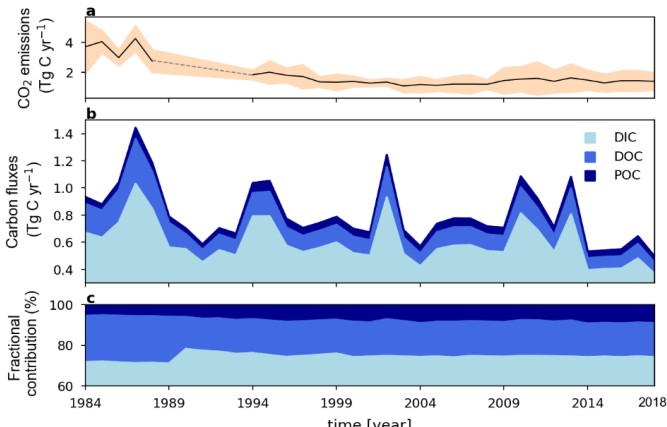

**Figure 4, (a) Temporal trends of yearly $CO_2$ emission rate from the Elbe River, dashed line indicates data gap; (b) Yearly dissolved inorganic carbon (DIC), dissolved organic carbon (DOC), and particulate organic carbon (POC) fluxes to the Elbe estuary; (c) Fractional contribution of each category to the total C flux.**

## 4    Discussion

### 4.1 Implications for $CO_2$ emissions in the Elbe River

The Elbe River has been a $CO_2$ source to the atmosphere most of the time, especially before 1990, and the $CO_2$ emission greatly reduced by water quality management. The annual mean $pCO_2$ before 1990 is like the one estimated during 1954–1977, demonstrating a substantial anthropogenic influence. This high $pCO_2$ was found to be significantly and positively correlated with TOC (Kempe, 1982). Additionally, the estimation of the Elbe estuary during the same period attributes the higher $pCO_2$ levels to the remineralization of organic matter (Amann et al., 2012). After 1990, a decrease in $pCO_2$ and $F_{CO2}$ was observed along with the improvement in water quality. Comparing with the world rivers, the latest $pCO_2$ in the Elbe River is slightly lower than in the estuary region (around 2500 µatm), and the global mean in-situ measure results (2705 µatm) (Amann et al., 2012; Liu et al., 2022), the wide range of $pCO_2$ values observed shows significant temporal variability in $CO_2$. The annual mean area specific $F_{CO2}$ in 2018 is also lower than that in the Elbe estuary and the annual mean of global rivers (Amann et al., 2014; Liu et al., 2022), indicate the substantial $CO_2$ exchange occurring between the river and the atmosphere.

Because the loads are strongly impacted by Q and thus exhibit seasonal characteristics, a seasonal decomposition for the monthly loads was applied to extract robust seasonal patterns (Fig. S3). The TN, TP, and DOC loads show relatively robust decreasing trends from the 1990s to 2018, indicating that water quality management reduces not only concentrations, but the loads input the coastal regions. By comparison, the POC and DIC concentrations and loads did not show significant trends although after the seasonal decomposition (Fig. S3). During the highly polluted period (before 1990), the seasonal trends of DOC were stronger than in other periods. Perhaps this is a clue to distinguishing significant anthropogenic influence and



natural processes. By comparison, the POC loads are more largely explained by flow discharge because of the additional
sources of terrestrial organic-rich soils, and sometimes one extreme event can contribute more than 90% of the organic carbon
loads in some semiarid regions (Ran et al., 2020).

The historical DIC, DOC, and POC fluxes enabled us to determine the amount of exported carbon from the entire Elbe
River basin. The highest proportion of carbon flux was organic carbon, both dissolved and particulate, with organic carbon
flux accounting for more than any other type of carbon. Additionally, vertical $CO_2$ fluxes accounted for a greater proportion
compared to horizontal input, indicating its integral role. Over time, there has been a gradual reduction in $CO_2$ emissions,
with emissions in 1990 more than halving by 2008. The most significant change occurred in the first 5 years, highlighting a
pronounced temporal variation in $CO_2$ emissions, especially under strong human influence. Therefore, the significant
temporal and seasonal variability highlights the importance of high-resolution data in improving estimation accuracy of
global carbon budget. Overall, our new dataset offers valuable insight into long time-series riverine $CO_2$ analysis.

### 4.1 Fluvial $CO_2$ emission reduction with water quality improvement

Both natural and human factors can impact the $CO_2$ efflux by altering $F_{CO2}$ and/or the water surface area. In the case of
$F_{CO2}$, $pCO_2$ is a more significant factor compared to $k_{600}$, accounting for almost 97% of the $F_{CO2}$ variations (Fig. 3f).
Additionally, the results indicate a general improvement in the water quality of the Elbe River from 1990 to 2018. While the
mainstem experienced more pronounced changes, improvements were also evident in the tributaries. These observations
underline the importance of managing nutrient inputs and maintaining oxygen levels in both the mainstem and tributaries of
river systems for recovery of eutrophication and $CO_2$ mitigation in the Elbe River. Therefore, the regression analysis of
environmental parameters and $pCO_2$ can help identify the contributing factors and quantify their effects on temporal variations
of $CO_2$ emissions in the Elbe River.

Organic matter originating from municipal wastewater directly discharged into the Elbe River may have been a crucial
contributor to $CO_2$ before 1990 (Amann, et al., 2012). This is because the aerobic respiration of this liable organic carbon
leads to the production of $CO_2$ and consumption of DO, contributing to an increase in $pCO_2$ in the river (Kerner and Yasseri,
1997; Amann et al., 2012; Kim et al., 2019), which also supported by the negative correlation between DO and $pCO_2$ ($r^2$=0.4,
p<0.01). Furthermore, during the lower DO condition, anaerobic carbon cycling might also act as an important source for not
only $CH_4$ and $N_2O$, but also for the $CO_2$ (Crawford et al., 2016). Moreover, colored substances from the eutrophication-
enhanced phytoplankton biomass present in the organic matter (OM) can reduce the efficiencies of photosynthesis occurring
in the water after a threshold, leading to an imbalance between photosynthesis and respiration, further elevated $CO_2$ levels
(Kim et al., 2021; Amann et al., 2012).

Another potential contributor to OM might be an increase in phytoplankton biomass due to eutrophication. This
eutrophication is typically stimulated by the rapid accumulation of human-made nutrients, such as those from urban and
farming runoff, because the alteration in N and P affects the quantity and variety of phytoplankton (Kothawala et al., 2021;
Jiang and Nakano, 2022). In its preliminary phase, the Elbe River was identified by a low TN/TP ratio of 37±13 before 1990,
which makes the conditions favorable for harmful algal blooms (HABs) to proliferate (Kim et al., 2019). However, the
negative correlation between $RUE_N$, $RUE_P$, and $pCO_2$ may suggest that under conditions of lower TN/TP, phytoplankton
exhibit a greater tendency to actively uptake $CO_2$ rather than engage in decomposition and respiration. The results indicate a
robust relationship, implying that the RUE is a superior predictor for riverine $pCO_2$ than merely the concentrations of TN and
TP (Figs. 3c, 3d).





The process of water quality treatment is key to reducing $pCO_2$ in the Elbe River. By removing organic matter from urban

wastewater before its discharge into the river, the amount of unstable organic carbon available for decomposition is diminished, thus leading to a subsequent drop in $CO_2$ generation. Furthermore, by decreasing the levels of TN and TP, balance in the aquatic ecosystem can be reestablished, thereby improving its capacity for $CO_2$ fixation. Following the dynamics outlined earlier, we constructed a regression model to forecast $pCO_2$ concentrations using inputs of $RUE_N$ and $RUE_P$ (equation 11, Fig. 5, $r^2$=0.51, p<0.01, n=1216). This equation highlights the relatively importance of $RUE_N$ (factor of -0.72) than the

$RUE_P$ (factor of -0.008), which might reveal the N might be the limiting nutrient of $pCO_2$ in the Elbe River.

$$lg_{pco2} = 3.24 - 0.72\ RUE_N - 0.008\ RUE_P \quad (11)$$

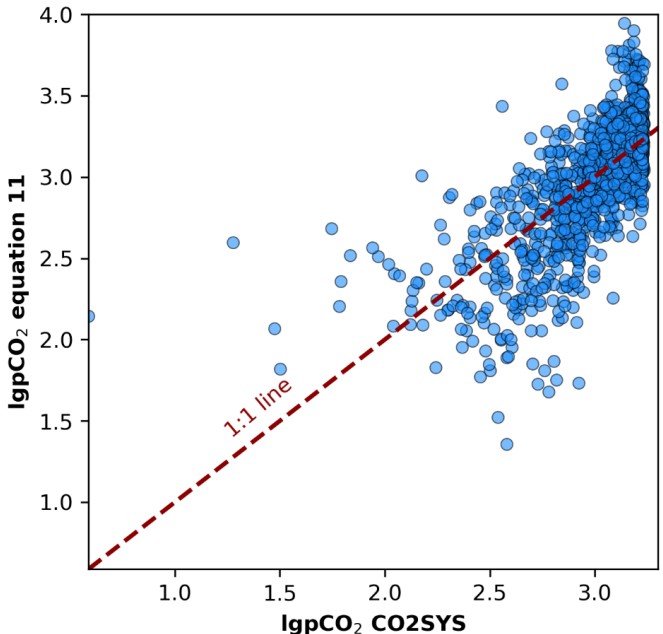

**Figure 5, Comparison of $lgpCO_2$ estimated by CO2SYS with modeled $lgpCO_2$ by equation 11 ($r^2$=0.51, p<0.01, n=1216)**

**4.3 The uncertainties of $CO_2$ drawdown with water quality improvement**

As discussed above, water quality management involving nutrients management and treatment could reduce fluvial $CO_2$ emission, especially in the first 10 years (Fig. 4a). The annual reduction rates were approximately 8.3 % until 1999, afterward the $F_{CO2}$ did not show a significant decreasing trend and kept fluctuating with the decreasing TN and TP concentrations from 2000–2018 (Fig. 1). This indicates the potential of $CO_2$ drawdown ratio by water quality management is limited. There are

two underlying factors that may account for this phenomenon. Firstly, the restoration of the ecosystem's capacity to support aquatic plants and their uptake of $CO_2$ is limited because the biomass amount could not increase in a restored aquatic system. Secondly, it is important to note that the control of OM, a significant contributor of $CO_2$ in natural systems, is challenging through water quality treatments. This is primarily because the input of OM before 1990s is mainly from anthropogenic sources, but afterward, the majority of POC is dominated by soil input, which is largely influenced by the natural hydrological



state. In some extraordinary cases, such as during floods, the annual flux of POC can be overwhelmingly dominated, accounting for 90% or more of the annual flux, which causes the uncertainties.

Besides, a proportion of $CO_2$ removal from the river might shift to $CO_2$ emission of wastewater treatment plants through biological treatment process and electricity consumption. According to global estimates, the degradation of OC during wastewater treatment in 2010 contributed to approximately 770 Tg $CO_2$-equivalent GHG emissions, represented nearly 1.57% of the total global GHG emissions of 49,000 Tg $CO_2$ (Edenhofer, 2015). On the other hand, the oxidized and anaerobic digestion of the organic carbon of wastewater is converted mainly to $CO_2$ and $CH_4$ (Campos et al., 2016), thus offsetting the reduction in $CO_2$ in waste water treatment. On the other hand, emerging technological advancements present promising solutions to address these issues effectively. For instance, in the realm of wastewater treatment plant operations, the adoption of renewable electricity as a replacement for traditional sources can yield substantial reductions in resulting emissions. Moreover, innovative approaches can be employed during water treatment processes to mitigate $CO_2$ generation. For example, clean carpet processes can replace the process of dephosphorization and denitrification, while the integration of alkalinity reactors could facilitate the capture of $CO_2$ throughout the treatment process (Lu et al., 2018). Overall, this study underscores the feasibility of implementing a well-managed strategy to curtail $CO_2$ levels in contaminated rivers, offering fresh support and inspiration for future carbon reduction endeavors.

**Conclusion**

This research demonstrates the significant reduction in $CO_2$ emissions in the Elbe River from 1984 to 2018, largely attributable to effective water quality management. Key factors contributing to these emissions include OC from municipal wastewater and an increase in phytoplankton biomass fueled by eutrophication. The management of nutrients of TN and TP has been shown to be crucial; it not only diminishes the sources of liable OC but also enhances the ecosystem's ability to absorb $CO_2$. Through the regression analysis of RUE and $pCO_2$, the study suggests that controlling N is more critical than P, and instead of contributing to decomposition and respiration, the abundance of phytoplankton actively participates in absorption of $CO_2$. However, the potential to reduce $CO_2$ emissions has its limitations, particularly due to unpredictable influxes of OC from natural occurrences. The effective use of wastewater treatment plants results in a relocation of $CO_2$ emissions from wastewater discharge to the river to emissions from wastewater treatment. However, emerging technologies may be able to mitigate such emissions.



**Data availability**

Data will be made available on a publicly available repository upon final publication.

**Author contributions**

MT, JH, and GM formulated the concept for this study. MT carried out the data analysis, receiving partial assistance from TA, LR, and JP. The text was written by MT, with contributions and assistance from all co-authors.

**Competing interests**

The authors have the following competing interests: Some authors are members of the editorial board of *Biogeosciences*.
The peer-review process was guided by an independent editor, and the authors have no other competing interests to declare.

**Acknowledgements**

The authors thank Matthias Wolf of the FGG Elbe for providing the Elbe monitoring data. This research was partly funded by the Deutsche Forschungsgemeinschaft (DFG, German Research Foundation) under Germany's Excellence Strategy - EXC
2037 'CLICCS - Climate, Climatic Change, and Society'- Project Number: 390683824, contributing to the Center for Earth System Research and Sustainability (CEN) of University of Hamburg. MT is funded by the China Scholarship Council (CSC).






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
