# Peer review of "Long-term reduction in CO2 emissions from the Elbe River due to"

_Biogeosciences, 2023_

## Author Comment (AC1)

Reviewer comments
Author responses
**RC1**
MAJOR COMMENTS
Long-term patterns of CO2 levels and emissions in rivers have been reported by several
studies (Jones et al. 2003; Ran et al. 2015; 2021; Nydahl et al. 2017; Marescaux et al.
2018) (non-exhaustive list). Findings from these studies could be used to contextualize the
present study (Introduction) and to discuss differences or convergences by comparison
(Discussion).
Reply:
Thank you for the references.
In the introduction, we will include a paragraph reviewing the literature on long-term $CO_2$
emission patterns. In the discussion, we will draw on previous studies to compare with our
findings and examine time-dependent variations in region-specific attributes.
L 30: « water pollution » is extremely vague. This should be broken down into several
human impacts on riverine systems that do not necessarily lead to the same change in
CO2 emissions. Eutrophication (increase of nutrient inputs) can potentially lead to
enhanced primary production and a CO2 sink in impounded large rivers such as the
Mississippi (Crawford et al. 2016). Conversely, croplands seem to also lead to enhanced
organic carbon inputs from soils enhancing CO2 emissions compared to more natural land
cover such as forests (Borges et al. 2018; Mwanake et al. 2023) Wastewater inputs lead
to CO2 production in the river, although this impact seems very local, in the near vicinity of
the emissary (Marescaux et al. 2018).
Reply:
Thank you. We agree with you that "water pollution" is indeed a broad term, and it is
important to consider its impacts on riverine $CO_2$ emissions from various perspectives.
Accordingly, we will expand our description to encompass different viewpoints, including
the effects of organic carbon from agricultural runoff and domestic sewage (Borges et al.,
2018; Marescaux et al., 2018; Mwanake et al., 2023), as well as the carbon sink impact
attributable to eutrophication caused by increased nutrient levels (Crawford et al., 2016).
L 30 "this percentage continues to increase because the unprecedented anthropogenic
stresses on riverine systems have led to many negative issues such as water pollution".
I'm not sure this statement applies assertively to all climate zones (Crawford et al., 2016).
According to Liu et al. (2022), tropical rivers are responsible for 57% of the riverine CO2
global emission, followed by temperate (30%) and Arctic regions (13%). The most direct anthropogenic impacts expected to affect riverine CO2 emissions should occur at
temperate latitudes (North America, Europe and parts of Asia) that account for less than a
third of total emissions. Note that this percentage was lower in earlier estimates for which
tropical rivers accounted for 80% of riverine CO2 emissions (Raymond et al. 2013;
Lauewarld et al. 2015).

Reply:
Thank you. We agree that the impacts of river pollution and restoration efforts on riverine
$CO_2$ emissions, which result from human activities, should be concentrated in regions with
high population density.

We will revise the sentence to offer a more accurate depiction that incorporates the
suggestions you have provided.

L 34: Rivers do not have "ecosystem's natural carbon absorption and storage capabilities".
Rivers do not store carbon in sediments and do not "absorb" carbon on contrary tend to
emit CO2 to the atmosphere. High CO2 over-saturation in rivers occurs ubiquitously even
in pristine (or near pristine) river basins such as the Amazon and Congo.

Reply:
Thank you very much for the correction. We agree that most rivers consistently serve as a
source of carbon.

We will revise the text from this perspective.

L 37: It has been argued that CO2 emissions from lowland rivers in particular in the tropics
are related to inputs from wetlands (Abril et al. 2014; Borges et al. 2015) that are
conceptually different (Abril and Borges 2019) from "terrestrial organic carbon (OC)» (as
stated).

Reply:
Thank you for the correction. We will add the reference and the information.

L38: Can you please clarify the role of «nutrient availability" in this context?

Reply:
We will delete the term "Nutrient availability" here, as it is misleading in this context.

L44-46: This argument is awkward. DOM produced by phytoplankton should indeed
sustain microbial respiration but phytoplankton also photosynthesized prior to DOM
release, so both effects should cancel each other in terms of net carbon fluxes.

Reply:
Thank you. The sentence will be deleted.

L 44: reference to "lakes and reservoirs » seems to be out of context here.

Reply:

Will be rephrased.

L49-50: statement "trophic status related to nutrient availability significantly impacts the levels of CO2 in rivers" is contradicted by the fact that CO2 emissions in rivers are in majority related to lateral inputs of carbon from soils and ground-waters (Hotchkiss et al.

2015) or from wetlands (Abril and Borges 2019), and are not related to in-stream CO2

production from metabolism (Hotchkiss et al. 2015; Abril et al. 2014; Borges et al. 2019).

Reply:

In this study, Figures 3c and 3d demonstrate the significant and negative correlation between RUE (the ratio of Chl-a to nutrient concentrations) and $pCO_2$.

The sentence will be rephrased deleting the terms 'trophic status related nutrient availability' and replaced by 'nutrient concentration'.

L 51: reference to "biodiversity" seems out of context here.

Reply:

Will be rephrased.

L 55: The authors should cite the "existing studies" they critique rather than stating this in a vague way.

Reply:

Thank you. Related studies will be cited (like Nydahl et al. (2017); Marescaux et al. (2018)

etc.)

L 55: Please clarify what is meant by "short term effects »? "effects" of what on what? Do you mean short-term time-series? Some studies have reported relatively long time series (Jones et al. 2003; Ran et al. 2015; 2021; Nydahl et al. 2017; Marescaux et al. 2018). It is not necessary to downplay existing literature to put forward your own study.

Reply:

Thank you. In our study, "Short-term effects" is a relative term compared with continuous long term time series, refers to the analysis of $F_{CO2}$ or $CO_2$ efflux below 10 years (decadal).

Will be clarified.

For the research you provided, While Ran et al. (2015) provided extensive data on long- term $pCO_2$, they did not conduct analyses related to $F_{CO2}$. After that, Ran et al. (2021)

compared $CO_2$ efflux from the average of two periods (1980s to the 2010s) but did not offer an exhaustive continuous time series analysis. Similarly, the work of Nydahl et al. (2017)

and Marescaux et al. (2018) was primarily directed towards understanding $pCO_2$ dynamics, with less emphasis on $F_{CO2}$. As a result, there is a research gap in continuous and long-term analyses of $F_{CO2}$ and $CO_2$ efflux, which our research questions aim to address. Will be rephrased and related studies will be included.

L 55: What do you mean by «hydrological conditions»? CO2 emissions from rivers depend on CO2 concentration between water and air, and on the gas transfer velocity. Both are more or less indirectly linked to "hydrological conditions" but this should be clarified, especially when criticizing "existing studies".

Reply:
Thank you. In this research, we are using estimates of both the flow discharge and flow velocity for the estimation of the gas transfer velocity and water surface area. The parameters represent hydrological conditions.

This aspect will be clarified in the text.

L61: Please provide a reference to back this statement, and clarify compared to which other rivers was it the most polluted? At European level? Globally? It could be also useful to take into account size effects. A very small stream can be extremely impacted by wastewater from a small village, while very large rivers are unaffected by large cities because all inputs are diluted by high discharge.

Reply:
Thank you. Before 1990, the Elbe River was one of the most polluted rivers in European scale. Related references will be added (ICPER, 2023; Kempe, 1982).

L 163: the equation relating river width and Q given by Raymond et al. (2012) was derived for small streams. Can you comment on its applicability to large rivers? Also this relation is probably affected by channelization and probably does not apply to highly engineered rivers such as the Elbe.

Reply:
Most of the Elbe River's flow, categorized with Strahler orders from 1 to 6, matches the flow discharge range used to create the equation by Raymond et al. (2012) (Figure R1).

[Figure]

Figure R1. Flow discharge distribution of tributaries of the Elbe River. Discharge data obtained and
resampled from GRADES (The Global Reach-scale A priori Discharge Estimates for SWOT) (Lin et al.,
2019; Yang et al., 2019).

For the larger segments of the river, classified as Strahler orders 7 and 8, primarily the
mainstem, we compared our estimated river widths with the research of Mallast et al.
(2020). Their measurements were derived from satellite imagery. The average river width
we estimated showed good agreement with their findings (this research: 177 m for Strahler
order 7&8 (Figure R2), versus Mallast et al. (2020): 183 m, with an area of 107 km² divided
by a length of 594 km).

Therefore, we believe the error introduced by our method in this research should be minor.
An additional discussion of uncertainties will be added.

[Figure]

Figure R2: Estimated River width across different Strahler orders.

L 300: can you please provide a numerical comparison and a reference for the data for
the 1954–1977 period?
Reply:
The modeled $pCO_2$ and the corresponding data and plot will be added.
Can you please explain somewhere in text why the analysis was not extended back to
1954 and only started in 1984?
Reply:
Since our study conducted a temporal and spatial analysis. In this process, we integrated
a range of environmental indicators along with carbon. On the other hand, the data from
1954 was restricted to just one site (sample location of the local water works company
Hamburg Wasser) and did not provide any environmental indicators (Kempe, 1982).
Consequently, we employed this data merely as a background reference value. A short
explanation will be added.
L341-344: This statement does not seem relevant. Indeed, it is conceivable that light
absorption by CDOM limits photosynthesis from aquatic primary producers, but in rivers
CDOM mostly originates from soils. Also, DOM from phytoplankton is usually very labile
and is quickly consumed by micro-organisms. CDOM is usually related to highly
refractory substances, typically from soils.
Reply:
The sentence will be deleted.
L 370-373: Please clarify the text of the two hypothesis and also provide extra arguments
and references to back them.
Reply:
The two main arguments are as follows:
Firstly, the treatment of municipal wastewater has resulted in a decrease in the amount of
labile organic carbon being directly introduced into the river, thereby reducing the potential
for its degradation into $CO_2$ (Lasaki et al., 2023). Secondly, the reduced discharge of heavy
metals, along with reductions in nitrogen and phosphorus concentrations, has promoted a
healthier aquatic ecosystem (Qasem et al., 2021). Although photosynthesis and respiration
processes may balance each other, the net growth of aquatic plants contributes to the
overall reduction of $CO_2$ in the river if the rate of plant growth exceeds the rate of
decomposition of plant residues (Demars et al., 2016).
Will be clarified with extra arguments and references.

What do you mean by "biomass amount » and why should it not increase in « restored
aquatic system"?
Reply:
The most important factor affecting biomass quantity is the toxicity from heavy metals,
which impedes biomass growth. As environmental conditions shift from polluted to non-
polluted states, the quantity of biomass is expected to change, subsequently influencing
$CO_2$ levels. However, heavy metals primarily originate from industrial inputs, and the
closure of factories along with advanced wastewater treatment technologies has
significantly improved water quality (Amann et al., 2012). Since trace elements do not
exceed the thresholds that limit phytoplankton growth, biomass remains relatively stable.
Will be rephrased. And a plot of temporal biomass amount variations will also be provided.
What do you mean by "challenging through water quality treatments."
Reply:
The challenge could be that $CO_2$ emissions from sewage water discharge may be avoided
at the cost of $CO_2$ emission of wastewater treatment plants through biological treatment
process and electricity consumption.
According to global estimates, the degradation of OC during wastewater treatment in 2010
contributed to approximately 770 Tg $CO_2$-equivalent GHG emissions, representing nearly
1.57% of the total global GHG emissions of 49,000 Tg $CO_2$ (Edenhofer, 2015).
On the other hand, the oxidized and anaerobic digestion of the organic carbon of
wastewater is converted mainly to $CO_2$ and $CH_4$ (Campos et al., 2016), thus offsetting the
reduction in $CO_2$ in wastewater treatment.
MINOR COMMENTS
Text contains numerous awkward phrasing or typos or redundancies. The senior co-
authors should spend some time looking through the text and make the necessary
improvements; this is not the reviewer's job. Nevertheless, some are listed hereafter (not
an exhaustive list):
Reply:
We apologize for this oversight. We will do our utmost to significantly improve the language
quality of the text.
L40 + L 337: Labile instead of "liable"?
Reply:
Will be replaced.

L 55: context instead of « contest" ?

Reply:
Will be replaced.

L42: "phytoplankton behaviors » is awkward, please rephrase.

Reply:
Will be rephrased.

L61: most instead of "highest"

Reply:
Will be replaced.

L66: "FCO2 efflux » is redundant sinc "F" of "FCO2" abbreviates the word flux.

Reply:
Will be replaced.

L68: "high-resolution" is self-evaluation, please simply state instead the actual time step of the data.

Reply:
Extra descriptions will be added.

L 368: "CO2 drawdown ratio by water quality management" is awkward, please rephrase.

Reply:
Will be rephrased.

**References**

**References**

Amann, T., Weiss, A., & Hartmann, J. (2012). Carbon dynamics in the freshwater part of the Elbe
estuary, Germany: Implications of improving water quality. *Estuarine, Coastal and Shelf*
*Science*, *107*, 112-121. https://doi.org/10.1016/j.ecss.2012.05.012

Borges, A. V., Darchambeau, F., Lambert, T., Bouillon, S., Morana, C., Brouyere, S., Hakoun, V.,
Jurado, A., Tseng, H. C., Descy, J. P., & Roland, F. A. E. (2018). Effects of agricultural land
use on fluvial carbon dioxide, methane and nitrous oxide concentrations in a large
European river, the Meuse (Belgium). *Sci Total Environ*, *610-611*, 342-355.
https://doi.org/10.1016/j.scitotenv.2017.08.047

Campos, J. L., Valenzuela-Heredia, D., Pedrouso, A., Val del Río, A., Belmonte, M., & Mosquera-
Corral, A. (2016). Greenhouse gases emissions from wastewater treatment plants:
minimization, treatment, and prevention. *Journal of Chemistry*, *2016*.

Crawford, J. T., Loken, L. C., Stanley, E. H., Stets, E. G., Dornblaser, M. M., & Striegl, R. G. (2016).
Basin scale controls on CO2 and CH4 emissions from the Upper Mississippi River.
*Geophysical Research Letters*, *43*(5), 1973-1979. https://doi.org/10.1002/2015gl067599

Demars, B. O. L., Gíslason, G. M., Ólafsson, J. S., Manson, J. R., Friberg, N., Hood, J. M., Thompson,
J. J. D., & Freitag, T. E. (2016). Impact of warming on CO2 emissions from streams
countered by aquatic photosynthesis. *Nature Geoscience*, *9*(10), 758-761.
https://doi.org/10.1038/ngeo2807

Edenhofer, O. (2015). *Climate change 2014: mitigation of climate change,* . Cambridge University
Press.

ICPER. (2023). International Commission for the Protection of the Elbe River · ICPER.
(https://www.ikse-mkol.org/ last access on 2023-12-04).

Kempe, S. (1982). Long-term records of CO2 pressure fluctuations in fresh waters. *SCOPE/UNEP*
*Sonderband*, *52*, 91-332.

Lasaki, B. A., Maurer, P., Schönberger, H., & Alvarez, E. P. (2023). Empowering municipal
wastewater treatment: Enhancing particulate organic carbon removal via chemical
advanced primary treatment. *Environmental Technology & Innovation*, *32*.
https://doi.org/10.1016/j.eti.2023.103436

Lin, P., Pan, M., Beck, H. E., Yang, Y., Yamazaki, D., Frasson, R., David, C. H., Durand, M., Pavelsky,
T. M., Allen, G. H., Gleason, C. J., & Wood, E. F. (2019). Global Reconstruction of
Naturalized River Flows at 2.94 Million Reaches. *Water Resour Res*, *55*(8), 6499-6516.
https://doi.org/10.1029/2019WR025287

Mallast, U., Staniek, M., & Koschorreck, M. (2020). Spatial upscaling of CO2 emissions from
exposed river sediments of the Elbe River during an extreme drought. *Ecohydrology*,
*13*(6). https://doi.org/10.1002/eco.2216

Marescaux, A., Thieu, V., Borges, A. V., & Garnier, J. (2018). Seasonal and spatial variability of the
partial pressure of carbon dioxide in the human-impacted Seine River in France. *Sci Rep*,
*8*(1), 13961. https://doi.org/10.1038/s41598-018-32332-2

Mwanake, R. M., Gettel, G. M., Wangari, E. G., Glaser, C., Houska, T., Breuer, L., Butterbach-Bahl,
K., & Kiese, R. (2023). Anthropogenic activities significantly increase annual greenhouse
gas (GHG) fluxes from temperate headwater streams in Germany. *Biogeosciences*,
*20*(16), 3395-3422. https://doi.org/10.5194/bg-20-3395-2023

Nydahl, A. C., Wallin, M. B., & Weyhenmeyer, G. A. (2017). No long-term trends in pCO2 despite
increasing organic carbon concentrations in boreal lakes, streams, and rivers. *Global*
*Biogeochemical Cycles*, *31*(6), 985-995. https://doi.org/10.1002/2016gb005539

Qasem, N. A. A., Mohammed, R. H., & Lawal, D. U. (2021). Removal of heavy metal ions from
wastewater: a comprehensive and critical review. *npj Clean Water*, *4*(1).
https://doi.org/10.1038/s41545-021-00127-0

Ran, L., Butman, D. E., Battin, T. J., Yang, X., Tian, M., Duvert, C., Hartmann, J., Geeraert, N., & Liu,
S. (2021). Substantial decrease in $CO_2$ emissions from Chinese inland waters due to
global change. *Nat Commun*, *12*(1), 1730. https://doi.org/10.1038/s41467-021-21926-6

Ran, L., Lu, X. X., Richey, J. E., Sun, H., Han, J., Yu, R., Liao, S., & Yi, Q. (2015). Long-term spatial
and temporal variation of CO2 partial pressure in the Yellow
River, China. *Biogeosciences*, *12*(4), 921-932. https://doi.org/10.5194/bg-12-921-2015

Raymond, P. A., Zappa, C. J., Butman, D., Bott, T. L., Potter, J., Mulholland, P., Laursen, A. E.,
McDowell, W. H., & Newbold, D. (2012). Scaling the gas transfer velocity and hydraulic
geometry in streams and small rivers. *Limnology and Oceanography: Fluids and*
*Environments*, *2*(1), 41-53. https://doi.org/10.1215/21573689-1597669

Yang, Y., Pan, M., Beck, H. E., Fisher, C. K., Beighley, R. E., Kao, S. C., Hong, Y., & Wood, E. F.
(2019). In Quest of Calibration Density and Consistency in Hydrologic Modeling:
Distributed Parameter Calibration against Streamflow Characteristics. *Water Resources*
*Research*, *55*(9), 7784-7803. https://doi.org/10.1029/2018wr024178

---

## Author Comment (AC2)

Reviewer comments
Author responses
**RC2**
This manuscript presents long-term water quality time series data from the Elbe River in
Europe. The authors use alkalinity and pH measurements to estimate dissolved CO2
concentrations, which they use to estimate CO2 emissions from the river and tributaries
from 1984 to 2018. They then compare the temporal changes in CO2 emissions with the
temporal changes in DIC, DOC and POC loads at the watershed's outlet, along with other
water quality parameters. The authors show a decrease in CO2 emissions, which they
relate to an improvement in water quality, particularly a decrease in DOC.
The paper suffers from several shortcomings in methodology, a poor presentation of
results, and considerable issues with the English language. I must admit this comes as a
surprise considering the list of authors, some of whom are widely recognized and
respected in the scientific community. I think there is potential to improve this paper
substantially, because the dataset holds significant value—but much more guidance will
need to be provided by the co-authors. In the following I will elaborate on the three main
concerns I have.
Reply:
Thank you for your thorough review. We apologize for shortcomings in the scientific
quality. We will do our best to revise the text, keeping in mind also language issues.
In the methodology section, we will include an analysis of the uncertainties associated
with $pCO_2$ and provided more details about the load calculations.
In the results section, we will incorporate a time series analysis of $pCO_2$ and biomass.
The results of the Mann-Kendall test will also be included.
**Methodological limitations**
One limitation is that the entire paper is based on the use of two indirect methods to
estimate CO2 emissions. First, pCO2 estimates are indirectly calculated from pH and
alkalinity measurements. While this is a common undertaking, the authors must at least
provide a quantification of uncertainties. Their plot comparing pCO2 estimates based on
two different packages (PHREEQC and CO2SYS) raises concerns as it shows large
differences between the two sets of estimates. Second, the CO2 emission estimates lack
actual measurements. The authors use an empirical model which was primarily
developed for smaller streams and might not be suitable to large rivers. The model
results are not evaluated against actual measurements. Again, this needs to be justified
(i.e. why was this particular model chosen and not another one?), and an assessment of
uncertainties should be presented.

Reply:
Thank you very much for your suggestions.
We selected CO2SYS (Lewis & Wallace, 1998) over Phreeqc (Parkhurst & Appelo, 2013)
for the calculations as not all datapoints provided the anions and cations required for a
reliable calculation in Phreeqc. About 60% of the sample points have major ion data. The
comparison of both calculations shows that there is an offset between measurements,
resulting in about 16% higher $pCO_2$ values (Figure S4), when calculated with CO2SYS.
To keep results consistent and comparable, in the study we calculated all data with
CO2SYS, accepting the potential error.
To evaluate the uncertainty in $pCO_2$ estimates, we will focus on the calculation methods,
as direct $pCO_2$ or $F_{CO2}$ measurements are unavailable. CO2SYS (Humphreys et al.,
2022) provides an approach to calculate the error propagation. The errors included in the
propagation are:
1) pH: General precision of standard commercial pH probes is typically between ±0.01 to
±0.1,so we will assume ±0.05.
2) TA: General precision of TA measurement by titration methods ranging from ±10 to
±50 µmol $L^{-1}$, so we will assume ±20 µmol $L^{-1}$.
3) Temperature: assumed as ±0.1 ℃.
Finally, this approach leads to an estimated uncertainty of around ±12% by CO2SYS.
Additionally, we will re-estimate the propagation errors in $CO_2$ efflux calculations using
the Monte Carlo method.
For the width estimation model by flow discharge from Raymond et al. (2012), which is
designed to the estimate of smaller rivers. For analysis the potential errors caused by this
equation. We compare our results from different Strahler orders:
Most of the Elbe River's flow, categorized with Strahler orders from 1 to 6, matches the
flow discharge range used to create the equation by Raymond et al. (2012) (Figure R1).

[Figure]

Figure R1. Flow discharge distribution of tributaries of the Elbe River. Flow discharge data obtained and resampled from GRADES (The Global Reach-scale A priori Discharge Estimates for SWOT) (Lin et al., 2019; Yang et al., 2019).

For the larger segments of the river, classified as Strahler orders 7 and 8, primarily the
mainstem, we compared our estimated river widths with the research of Mallast et al.
(2020). Their measurements were derived from satellite imagery. The average river width
we estimated showed good agreement with their findings (this research: 177 m for
Strahler order 7&8 (Figure R2), versus Mallast et al. (2020): 183 m, with an area of 107
km² divided by a length of 594 km).
Therefore, we believe the error introduced by our method in this research should be
minor. An additional discussion of uncertainties will be added.

[Figure]

Figure R2: Estimated River width across different Strahler orders.

Another critical issue is with the use of discharge values for k600 estimates. From what I
gather, the authors have used only one discharge value for each river location. This
approach is problematic because k600 is highly influenced by discharge fluctuations, and
failing to account for discharge fluctuations will result in erroneous CO2 emission flux
estimates. This issue becomes evident in Figure 3f, where the relationship between
FCO2 and pCO2 is almost perfectly linear—either suggesting that k600 has no influence
on FCO2, or that k600 remains constant across space and time, both of which are
improbable.

Reply:
In our study, the flow data used for calculations were extracted from the GRADES
database (Lin et al., 2019; Yang et al., 2019), specifically selected to correspond with the
dates of hydro-chemical data sampling. This database offers daily records of flow
discharge, inherently accounting for the influence of flow variations on the seasonal $k_{600}$
values. We will also provide the correlation analysis between variations in $k_{600}$ and $F_{CO2}$.
Additionally, a comparative analysis between data from the GRADES database and
actual measurements provided by hydrological stations will be conducted. A short
discussion in long-term changes in discharge and the impact on $F_{CO2}$ will be included.
A third issue is with the DOC data. It appears that two methods are used for the DOC flux
estimation, yet only one is presented in the Results section. Furthermore, the first method
does not present a way to calculate loads, but simply provides a framework for
classifying C-Q patterns, which is rather confusing.
Reply:
Two calculation methods are described in the text, both founded on the principle of fitting
the concentration to a model that utilizes the flow discharge to adjust the concentration.
These approaches result in final errors that stem from the differences between the
measured values and the values derived from model fitting.
Detailed explanations of this calculation process and uncertainties analysis will be
provided in the methods section and in the supplementary.
Furthermore, upon comparison, the results from two methods show little differences.
Therefore, we have applied the average of the two as the result.
**Presentation of results**
The results of statistical tests are not consistently reported throughout the paper. For
example, Mann-Kendall trend test results are not presented for pCO2 and FCO2 (L231-
261) as well as for DIC, DOC and POC (L276-291), making it challenging to assess the
significance of the observed trends. Furthermore, there are no reported step change test
results, despite the mention of these tests in the Methods section.

Reply:
The results of the Mann-Kendall test and the step change test for parameters such as $pCO_2$, $F_{CO2}$, DIC, DOC, and POC, etc., will be added.

I also noted some inconsistent statements between the results and discussion: while on L281 the authors state that "POC, DOC and DIC loads did not show significant trends", this contradicts the following statement that the DOC load "showed relatively robust decreasing trend" (L310-311).

Reply:
According to the Mann-Kendall test, DOC load exhibit significant decreasing trends ($p < 0.01$).

We will address this correction and thoroughly review the entire manuscript to avoid such mistakes.

Lastly, several figures are missing. For instance, the pCO2 time-series data are not shown despite these data being arguably one of the most critical data of the paper.

Reply:
A set of more comprehensive analyses, including additional time-series plots with $pCO_2$ and other parameters, will be added to the manuscript.

**English language**
The paper is very challenging to understand, and clearly the more senior authors (some of whom are well-published) have not provided the necessary feedback. It seems like only the abstract and the first few paragraphs of the introduction have been edited. The language used is awkward at best, and completely incoherent at worst. As a reviewer, I am not willing to invest one or two days correcting grammar and editing the entire paper. I strongly recommend that the senior authors fulfil their responsibilities of reviewing and editing this paper.

Reply:
We will fully revise the manuscript from all the co-authors.

**References**

Humphreys, M. P., Lewis, E. R., Sharp, J. D., & Pierrot, D. (2022). PyCO2SYS v1.8: marine carbonate system calculations in Python. *Geosci. Model Dev.*, *15*(1), 15-43. https://doi.org/10.5194/gmd-15-15-2022

Lewis, E., & Wallace, D. (1998). *Program developed for CO2 system calculations*.

Lin, P., Pan, M., Beck, H. E., Yang, Y., Yamazaki, D., Frasson, R., David, C. H., Durand, M., Pavelsky, T. M., Allen, G. H., Gleason, C. J., & Wood, E. F. (2019). Global Reconstruction of Naturalized River Flows at 2.94 Million Reaches. *Water Resour Res*, *55*(8), 6499-6516. https://doi.org/10.1029/2019WR025287

Mallast, U., Staniek, M., & Koschorreck, M. (2020). Spatial upscaling of CO2 emissions from exposed river sediments of the Elbe River during an extreme drought. *Ecohydrology*, *13*(6). https://doi.org/10.1002/eco.2216

Parkhurst, D. L., & Appelo, C. (2013). Description of input and examples for PHREEQC version 3—a computer program for speciation, batch-reaction, one-dimensional transport, and inverse geochemical calculations. *US geological survey techniques and methods*, *6*(A43), 497.

Raymond, P. A., Zappa, C. J., Butman, D., Bott, T. L., Potter, J., Mulholland, P., Laursen, A. E., McDowell, W. H., & Newbold, D. (2012). Scaling the gas transfer velocity and hydraulic geometry in streams and small rivers. *Limnology and Oceanography: Fluids and Environments*, *2*(1), 41-53. https://doi.org/10.1215/21573689-1597669

Yang, Y., Pan, M., Beck, H. E., Fisher, C. K., Beighley, R. E., Kao, S. C., Hong, Y., & Wood, E. F. (2019). In Quest of Calibration Density and Consistency in Hydrologic Modeling: Distributed Parameter Calibration against Streamflow Characteristics. *Water Resources Research*, *55*(9), 7784-7803. https://doi.org/10.1029/2018wr024178

---

## Author Comment (AC3)

Reviewer comments
Author responses
**CC1**
It is a very good idea to use long term monitoring data to investigate the effect of the
socioeconomic changes in Germany after re-unification on GHG emissions from a large
river. The paper contains a very nice dataset including both main river and tributary data
which allows the investigation of both spatial and inter-annual pattern. However, in my eyes
the manuscript does not fully exploit the potential of the dataset and has some serious
issues which I would like to address in the following:
Reply:
Thank you for your thorough review and valuable additional input.
I cannot follow the argumentation that nutrient driven eutrophication should increase $CO_2$.
Any $CO_2$ produced from decaying algae was fixed by those algae before. Thus, the cycle
of primary production and algae mineralization cannot increase $CO_2$ emissions. In contrast
it has the potential to reduce $CO_2$ emissions if algae are buried in the sediments – a
scenario relevant for lakes but probably not for rivers.
Reply:
Thank you. It's true that eutrophication typically results in a decrease of $CO_2$ due to the
uptake by photosynthetic phytoplankton. However, recent research suggests this impact
could be reversed. For instance, Kim et al. (2021) found a V-shaped relationship between
TN/TP and $pCO_2$, together with upshift relation between Chl-a and $CO_2$, indicating that
beyond a certain threshold, eutrophication enhanced biomass could act as a source of $CO_2$
in the Han River, Korea. This is why we initially highlighted this potential. Ultimately, our
results for the Elbe demonstrated a negative relationship between biomass and $pCO_2$
(Figures 3c and 3d), indicating effects of uptake rather than impact as a source in the Elbe
River.
Will be rephrased.
I would hypothesize that correlation between N or P with $CO_2$ might be a pseudo correlation
and not a direct mechanistic link. As written in the manuscript, wastewater contains both
DOC and inorganic nutrients.
Reply:
In fact, in our view we believe that if the decrease in $CO_2$ were solely correlated with the
amount of organic carbon, then a direct correlation between $pCO_2$ and DOC or TOC would
be expected, which was not identified. The negative correlation between RUE and the
$pCO_2$ suggests that biomass carbon uptake efficiency has the potential to contribute to the
decrease in $pCO_2$.

In addition, before the unification, less wastewater treatment in Eastern Germany led to
higher more labile carbon input to the river water.
In the manuscript a rather crude method is used to estimate river surface area. The
resulting surface area of 735 km$^2$ (supplement) looks rather high. Divided by river length
this means a river width of about 1 km – an unrealistic high value. In Mallast et al. (2020)
we determined a surface area of 106 km$^2$ from satellite images.
Reply:
Thank you for providing this valuable reference information.
Mallast et al. (2020) utilized high-resolution satellite imagery to estimate the water area
with great accuracy, which only included mainstem portion, refer to the 7-8 Strahler order
river network considered for our estimation.
We also extracted this segment of the river network for comparison. Based on our
estimation results for 2018, the river width results are quite similar, (this research: 177 m
for Strahler order 7&8, versus Mallast et al. (2020): 183 m, with an area of 107 km² divided
by a length of 594 km) (Figure R1).
Therefore, we believe that the error in our estimation is not significant, and the results are
reliable. And we will add the comparation results for the uncertainties discussion.

Figure R1: Estimated River width across different Strahler orders.
The gas transfer velocity was estimated from slope and flow velocity. However, there are
also k600 data from River Elbe published (Matoušů et al., 2019). It should at least be
checked how estimated k600 data compare to measured ones.
Reply:
We will compare the results with our calculations.

In Kamjunke et al. (2022) and Kamjunke et al. (2023) it was shown that there is a longitudinal gradient with plankton concentrations increasing downstream the river. It would be interesting to analyze the dataset in this paper with respect to this gradient. Was the transition zone between plankton poor and plankton rich water moving downstream after 1990?

Reply:
We are also interested in examining the longitudinal gradient in relation to plankton concentrations to identify the transition zone between areas of plankton-poor and plankton-rich waters downstream after 1990.

We will add data concerning the longitudinal gradient to the plankton concentrations from a series of monitoring stations in case they show meaningful results.

The dataset also should allow the comparison of different tributaries. Statistical relations between $CO_2$ and other parameters could be checked for each tributary separately. This can be used to investigate the drivers of $CO_2$ in the different sub-catchments. The effect of the tributaries on the main stream, however, is probably difficult to detect. In Bussmann et al. (2022) for example we showed that the high dilution effect at the confluence did not allow the detection of $CH_4$ import from the tributaries into the main river.

Reply:
Thank you for your valuable suggestion to utilize the dataset for comparing various tributaries and examining the statistical relationships between $CO_2$ and other parameters for each tributary individually. We will attempt to conduct separate statistical analyses for each tributary where water chemistry data are available.

Recent literature shows that $CO_2$ concentrations in rivers fluctuate diurnally (Gómez-Gener et al., 2021). Thus, scaling up $CO_2$ emissions from single datapoints means accepting a systematic uncertainty. Our own measurements show that diurnal fluctuation of $CO_2$ is an issue in River Elbe (manuscript in preparation). This could be relevant in long term time series, if the time of day when samples were taken changed during the time series.

Reply:
Thank you. The long-term effect is also important since respiration obviously dominates photosynthesis during the night (Gómez-Gener et al., 2021). However, according to the datasets of FGG, the time distribution listed below (Figure R2), most of the sampling happened during daytime, therefore, it is not available to analysis the impact in this event.

[Figure]

Figure R2. Distribution of manual sampling times in the Elbe River

If $CO_2$ emissions are primarily driven by DOC mineralization the dataset should allow a
quantitative comparison between the two. Was DOC decreasing downstream and how
does that downstream decrease of DOC compare quantitatively to $CO_2$ emissions? Such
a question could be investigated by looking at monitoring data from longer reaches
without major tributaries.

Reply:
Indeed, decreasing trends for both DOC fluxes and $CO_2$ emissions have been observed
in our research. Conducting a quantitative analysis to determine whether the decrease in
DOC is the primary driver of the reduction in $CO_2$ emissions in the Elbe River is an
excellent suggestion.

We will add a section on quantitative analysis to address this.

An analysis of long-term changes of water quality in river Elbe was recently published by
(Wachholz et al., 2022)

Reply:
The recent publication highlights the decrease in nutrient concentrations and
underscores the importance of water quality management, providing new resources for
our research.

We will add it as a reference.

**References**

Gómez-Gener, L., Rocher-Ros, G., Battin, T., Cohen, M. J., Dalmagro, H. J., Dinsmore, K. J., Drake, T. W., Duvert, C., Enrich-Prast, A., Horgby, Å., Johnson, M. S., Kirk, L., Machado-Silva, F., Marzolf, N. S., McDowell, M. J., McDowell, W. H., Miettinen, H., Ojala, A. K., Peter, H., . . . Sponseller, R. A. (2021). Global carbon dioxide efflux from rivers enhanced by high nocturnal emissions. *Nature Geoscience*, *14*(5), 289-294. https://doi.org/10.1038/s41561-021-00722-3

Kim, D., Lim, J. H., Chun, Y., Nayna, O. K., Begum, M. S., & Park, J. H. (2021). Phytoplankton nutrient use and CO(2) dynamics responding to long-term changes in riverine N and P availability. *Water Res*, *203*, 117510. https://doi.org/10.1016/j.watres.2021.117510

Mallast, U., Staniek, M., & Koschorreck, M. (2020). Spatial upscaling of CO2 emissions from exposed river sediments of the Elbe River during an extreme drought. *Ecohydrology*, *13*(6). https://doi.org/10.1002/eco.2216